# Effect of Elicitation with (+)-Usnic Acid on Accumulation of Phenolic Acids and Flavonoids in Agitated Microshoots of *Eryngium alpinum* L.

**DOI:** 10.3390/molecules26185532

**Published:** 2021-09-12

**Authors:** Małgorzata Kikowska, Barbara Thiem, Karolina Jafernik, Marta Klimek-Szczykutowicz, Elżbieta Studzińska-Sroka, Halina Ekiert, Agnieszka Szopa

**Affiliations:** 1Department of Pharmaceutical Botany and Plant Biotechnology, University of Medical Sciences in Poznan, 14 Św. Marii Magdaleny St., 61-861 Poznań, Poland; bthiem@ump.edu.pl; 2Department of Pharmaceutical Botany, Collegium Medicum, Jagiellonian University, 9 Medyczna St., 30-688 Kraków, Poland; karolina.jafernik@doctoral.uj.edu.pl (K.J.); marta.klimek-szczykutowicz@doctoral.uj.edu.pl (M.K.-S.); halina.ekiert@uj.edu.pl (H.E.); a.szopa@uj.edu.pl (A.S.); 3Department of Pharmacognosy, University of Medical Sciences in Poznan, 4 Święcickiego St., 61-781 Poznań, Poland; elastudzinska@ump.edu.pl

**Keywords:** alpine eryngo, microshoot cultures, phenolic compounds, elicitation, HPLC-DAD analysis

## Abstract

The present work was aimed at studying the potential of elicitation on the accumulation of phenolic compounds in in vitro shoot cultures of *Eryngium alpinum* L., a protected plant from the Apiaceae family. The study examined the influence of (+)-usnic acid on the biomass growth as well as on the biosynthesis of the desired flavonoids and phenolic acids in the cultured microshoots. The phenolic compound content was determined by HPLC-DAD. The flavonoid of the highest concentration was isoquercetin, and the phenolic acids of the highest amount were rosmarinic acid, caffeic acid and 3,4-dihydroxyphenylacetic acid, both in the non-elicited and elicited biomass. Isoquercetin accumulation was efficiently increased by a longer elicitation with a lower concentration of lichenic compound (107.17 ± 4.67 mg/100 g DW) or a shorter elicitation with a higher concentration of acid (127.54 ± 11.34 and 108.37 ± 12.1 mg/100 g DW). Rosmarinic acid production generally remained high in all elicited and non-elicited microshoots. The highest content of this acid was recorded at 24 h of elicitation with 3.125 µM usnic acid (512.69 ± 4.89 mg/100 g DW). The process of elicitation with (+)-usnic acid, a well-known lichenic compound with allelopathic nature, may therefore be an effective technique of enhancing phenolic compound accumulation in alpine eryngo microshoot biomass.

## 1. Introduction

*Eryngium alpinum* L. is an herbaceous perennial plant from the Apiaceae family. The International Union for Conservation of Nature’s Red List of Threatened Species indicates that this species is ‘Near Threatened’ [1]. According to the European Environment Agency, it is protected by EU Habitats Directive, Bern Directive, and Natura 2000 [2]. In accordance with the Catalogue of Life, it is native to Austria, Liechtenstein, France, Switzerland, Italy, Bosnia and Herzegovina, Montenegro, and Croatia [3]. Wild populations are in decline due to problems with generative reproduction, strong dormancy of seeds and poor germination, overcollection for ornamental purposes, and habitat degradation [4]. 

Due to the environmental and climatic changes affecting the fragmentation of the habitats and the protection status of *E. alpinum*, it is not possible to harvest the material from the natural environment. For this reason, in vitro systems may become an alternative source of the genetically aligned plant biomass. Nowadays, plant in vitro systems bring many advantages—they enable continuous production of uniform biomass from protected species independently of climatic, environmental and soil conditions. Moreover, plant biomass with good biotechnological parameters may become a material for phytochemical and biological research, without the overexploitation of the natural environment [5]. An important advantage of agitated microshoot in vitro cultures is the possibility of affecting the accumulation of desired phenolic compounds in biomass, applying an elicitation. It is an innovative research system that enables the study of organ biology and the possibility of accumulation of selected compounds under the application of biotechnology-based approaches [6].

Phenolic acids, flavonoids, coumarins and other small molecules of phenolic compounds, as well as triterpenoid saponins and constituents of essential oil, are the main secondary metabolites in the herb of *E. alpinum*. Phenolic acids (chlorogenic acid, rosmarinic acid, and its derivative R-(+)-3′*O*-*β*-D-glucopyranosyl rosmarinic acid [7], as well as caftaric acid, neochlorogenic acid, isochlorogenic acid, 3,4-dihydroxyphenylacetic acid, and caffeic acid [8,9]) have previously been found in this species. Moreover, the group of hydroxycinnamic acid derivates: coumaroylcholine, caffeoylcholine, feruloylquinic acid, dicaffeoylquinic acid, and ferulic acid, and methyl rosmarinate, as well as benzoic acid derivates: glucosyringic acid, trimethoxybenzoic acid, vanillic acid, hydroxyphenyllactic acid, dimethoxybenzaldehyde, glucovanillin, have also been found [10]. Additionally, flavonoids (quercetin and kaempferol [11], isoquercetin and quercitrin [8,9], luteolin, luteolin 7-*O*-glucoside, kaempferol-3-*O*-rhamnoside, kaempferol-3-*O*-rutinoside, quercetin-3-*O*-glucoside, quercetin-3-*O*-galactoside, quercetin-3-*O*-rutinoside [10]) and coumarins (umbelliferone, scopoletin, 4-methylumbelliferyl glucuronide, fraxidin, isofraxidin, scopoline, esculin, dihydroxycoumarin) have been found in this species [10]. The metabolites detected were triterpenoid saponins (3-*O*–*β*-D-glucopyranosyl-(1→2)-*β*-D-glucuronopyranosyl-22-*O*-angeloyl-R1-barrigenol, 3-*O*-*β*-D-glucopyranosyl-(1→2)-*β*-D-glucuronopyranosyl-22-*O*-angeloyl-A1-barrigenol) and eryngioside J [10]. The main constituents identified by GC-FID and GC/MS in the essential oil of *E. alpinum* isolated by hydrodistillation of the aerial parts of the plant were caryophyllene oxide and *α*-bisabolol from oxygenated sesquiterpenes, as well as bicyclogermecrene and germacrene D [12]. Recent analysis of this species has shown the presence of *β*-elemenone, germacrone, two selinadienes, and 1,8-cineole in the leaf oil as well as hexadecanoic acid, spathulenol, (*E*)-*β*-farnesene, and falcarinol in the in vitro shoot oil [13].

The expansion of the little knowledge in the field of phytochemistry of this endangered species was possible thanks to the multiplication of the microshoot biomass in in vitro cultures. The previous articles on phytochemical investigation of bioactive compounds in *E. alpinum* tissues demonstrated that the biomass from microshoot cultures is able to produce comparable or even much higher amounts of phenolic compounds to the shoots of soil grown plants. The results of previous studies prompted the authors of the experiments to undertake efforts to increase the content of bioactive compounds in the cultured biomass. The review of the literature on biotechnological procedures concerned the use of mainly UV radiation, fungal extracts (e.g., yeast extract), hyperosmotic stress, heavy metals or methyl jasmonate to increase the content of secondary metabolites. New compounds of natural origin that could play the role of elicitors are still being searched. Usnic acid, a compound present in the lichen thallus, was selected for this search as an elicitor influencing the accumulation of selected phenolic compounds. Neither the use of (+)-usnic acid for elicitation of tissues of soil-grown plants, nor plant biomass from in vitro cultures, has been studied so far.

About 17,000 lichen species are known and new species are discovered every year. Lichens are one of the most important sources of biologically active compounds, but the biological role of lichen substances are still at a speculative level. On the basis of the existing scientific literature, usnic acid is one of the most common exclusive lichenic metabolites. The usnic acid produced by the fungal partner is a well-known natural herbicide, however the information about its physiological role is scanty. Lichens have to compete against mosses and liverworts in many habitats, and their chemical capacity of suppressing these organisms could be of precious adaptive value. Moreover, usnic acid is an allelopathic agent, inhibiting moss spore germ. Studies supporting the allelopathic activity of lichen substances on vascular plants have been carried out by several authors, but the adaptive value of this inhibiting effect is still difficult to assess [14,15,16]. Due to the allelopathic nature of usnic acid in relation to fungi or plants, an attempt was made to investigate the possibility of playing the role of this lichen acid as an elicitor in relation to plant cells.

This work investigated the effect of the elicitation with usnic acid on the accumulation of phenolic compounds in alpine eryngo microshoot cultures. Phenolic compounds are a large group of plant bioactive compounds showing a diversity of structures. Phenolic acids are most commonly hydroxy or methoxyl derivatives of benzoic and *trans*-cinnamic acid, often occurring in the form of ester or ether connections with other acids, however glycoside derivatives are also commonly found. Flavonoids are polycyclic compounds based on the phenylbenzo-*γ*-chromium skeleton substituted with hydroxyl groups in different positions [17]. The variety of health-promoting benefits of this class of secondary metabolites include, e.g., antioxidant, antimicrobial, anti-inflammatory, antimutagenic, anticancer, antihypertensive, diuretic, hepatoprotective, and vasoprotective effects [18]. The production of phenolic compounds in in vitro plant cultures was more widely discussed in a review [19].

The aim of this study was to obtain the microshoot biomass of *E. alpinum* under in vitro conditions and to comprehensively analyze the quality and quantity of phenolic acids and flavonoids before and after elicitation with (+)-usnic acid. The shoots were multiplied by means of the axillary bud proliferation technique in agitated liquid media. The present work examined the influence of (+)-usnic acid on the biomass growth as well as on the accumulation of the studied phenolic compounds in the cultured microshoots.

This work presents the results of the study aimed at examining the influence of an elicitation in the stabilized microshoot cultures in the previously selected medium. Due to the abundance of works on the elicitation of plant cells with biotic elicitors such as yeast extract and fungi post-culture media, the lichenic compound (+)-usnic acid was used in this experiment.

## 2. Results

Taking into account the results of previous experiments [8,9] of the selection of in vitro conditions in which the tissues of *E. alpinum* shoots accumulated an increased content of bioactive phenolic compounds, the best composition of the medium and the system of agitated shoots were selected for the use of biotechnological treatment.

### 2.1. Stabilization of Shoot Cultures Agitated in Liquid Media

Before applying elicitation, the microshoot culture of *E. alpinum* was stabilized under controlled conditions. For this reason, the number and length of cultured microshoots were measured (Table 1) and the biomass quality was assessed (Figure 1). 

Agitated shoots from liquid media, both non-elicited and elicited, were characterized by intense green color and correct morphology (Figure 1). The influence of various concentrations of gibberellic acid on shoot number and length was investigated in this study. 

As shown in Table 1, the addition of growth regulators (PGRs) to the liquid media significantly influenced the number of new shoots produced compared to the control medium without supplementation with phytohormones. PGRs (benzylaminopurine (BAP), indolile-3-acetic acid (IAA) and gibberellic acid (GA_3_)) greatly influenced the formation of new shoots. However, the number of new shoots was not statistically different depending on the PGR composition or GA_3_ concentration (7.66 ± 0.58–9.33 ± 1.15 shoots per explant compared to the control (2.67 ± 0.58 shoots per explant)). Moreover, GA_3_ affected the shoot length (7.60 ± 0.66–8.03 ± 1.17) but this parameter was not statistically different depending on the gibberellin concentration used. For this reason, using the lowest concentration of this PGR is justified (Table 1).

### 2.2. Effect of (+)-Usnic Acid on Phenolic Compound Content in Shoot Cultures 

This study examined the effect of 12, 24, 48 and 72 h elicitation with (+)-usnic acid (3.125–50 µM) on phenolic compounds (phenolic acids and flavonoids) in *E. alpinum* microshoots cultured in vitro. The identification of phenolic compounds was performed using the HPLC-DAD method to compare their retention times with standards.

The qualitative analysis shows that 3 flavonoids (isoquercetin, robinin, quercitrin) out of 14 used standards (Table 2) and 7 phenolic acids (caftaric, 3,4-dihydroxyphenylacetic acid, chlorogenic, cryptochlorogenic, isochlorogenic, caffeic, rosmarinic acids) out of 23 tested compounds were detected in methanol extracts. 

As shown in Table 2 and Table 3, 10 phenolic compounds were detected in methanol extracts from elicited shoot biomass of *E. alpinum* (Table 3). In most cases, the content of selected flavonoids and phenolic acids in biomass depended on the concentration of the elicitor and the exposure time of the microshoots to (+)-usnic acid. Although, in a very simplified way, it can be said that the content of flavonoids in the microshoot biomass varied to a lesser extent depending on the parameters of the elicitor used rather than the level of phenolic acids (Table 2 and Table 3).

Among flavonoids, the isoquercetin accumulation was most efficiently determined by longer elicitation with a lower concentration of lichenic compound (72 h with 3.125 µM = 107.17 ± 4.67 mg/100 g DW) or shorter elicitation with a higher concentration of acid (12 h with 12.5 µM or 25 µM = 127.54 ± 11.34 and 108.37 ± 12.1 mg/100 g DW). 

In the case of robinin content, the value was at a similar level in elicited microshoots as the non-elicited microshoots, and the significant decrease of this particular flavonoid compound was observed after longer time of treatment. The elicitation with highest concentration of (+)-usnic acid resulted in the lowest amount of robinin in microshoot biomass. The highest content of robinin was achieved after addition of 3.125 µM usnic acid to the culture medium for 24 h (37.32 ± 2.59 mg/100 g DW).

In general, the content of quercitrin in microshoots was similar regardless of the concentration of the elicitor or the treatment time. However, in the case of the lowest concentration of usnic acid (3.125 µM), the content of this flavonoid clearly increased regardless of the exposure time, reaching the highest value after 72 h elicitation (51.08 ± 0.17 mg/100 g DW) (Table 2).

As shown in Table 3, the elicitation with (+)-usnic acid increased the content of 3,4-dihydroxyphenylacetic acid in almost all cases. The highest content of this phenolic acid was detected in microshoots of *E. alpinum* elicited with 25 µM usnic acid for 12 h (205.47 ± 17.02 mg/100 g DW).

On the other hand, no increase in content of cryptochlorogenic and isochlorogenic acids was observed in microshoot biomass after the elicitation process. 

For caftaric acid, the treatment of cultured shoots with higher concentrations (6.25 µM, 12.5 µM and 25 µM) of elicitor for 12 h and 24 h or a lower concentration (3.125 µM) of elicitor for 24 h/72 h was beneficial for its accumulation. The highest content of this phenolic acid was detected in microshoots elicited with 3.125 µM usnic acid for 24 h (43.32 ± 1.11 mg/100 g DW). 

In general, for chlorogenic acid accumulation, longer exposure times to the lichenic compound used at a higher concentration proved to be successful. The highest contents of this phenolic acid were detected in microshoots elicited with 50 µM usnic acid for 48 h (43.44 ± 0.33 mg/100 g DW), 25 µM usnic acid for 48 h (32.47 ± 1.48 mg/100 g DW), and 12.5 µM usnic acid for 48 h (31.29 ± 1.09 mg/100 g DW). 

In addition, a prolongation of elicitation time had a positive effect on the biosynthesis capacity of caffeic acid. The greatest contents of this phenolic acid were detected in microshoots elicited with 50 µM usnic acid for 48 h (218.38 ± 0.95 mg/100 g DW) and 25 µM/ 12.5 µM / 3.125 µM usnic acid for 72 h (217.78 ± 1.84, 175.51 ± 9.26, 178.93 ± 7.41 mg/100 g DW, respectively). 

Rosmarinic acid production generally remained high in all elicited and non-elicited microshoots. The maximum yield of this acid was recorded at 24 h of elicitation with 3.125 µM usnic acid (512.69 ± 4.89 mg/100 g DW). However, high amounts were also observed when eliciting with higher concentrations of the elicitor (Table 3).

## 3. Discussion

*E. alpinum* was introduced to in vitro cultures and the capacity to produce the desired phenolic compounds (phenolic acids and flavonoids, as well as essential oil) by the microshoot biomass under controlled conditions was previously investigated [8,9,10]. The accumulation of phenolic compounds in the shoot biomass from in vitro cultures grown in different systems and on different media was determined, compared to their content in the organs of soil-grown plants [8,9].

In this study, agitated shoots from liquid media, both non-elicited and elicited, were characterized by intense green color and correct morphology. However, in the previous studies, microshoots of *E. alpinum* agitated in liquid media were characterized by their abnormality and hyperhydricity [8]. We observed that this may be the effect of high medium volume in the culture vessel (30 mL), and therefore its volume was reduced to 20 mL in the present experiment, which positively influenced the quality of the microshoots. Earlier experiments [8,9] investigated the effect of media supplementation with various concentrations of plant growth regulators: benzylaminopurine (BAP), indolile-3-acetic acid (IAA) and gibberellic acid (GA_3_) on the growth of *E. alpinum* shoot biomass in various culture systems [8,9]. Since the selection of plant growth regulators (PGRs: BAP + IAA + GA_3_) did not affect the statistical differences in the number of multiplied shoots [8,9], and it is known from the scientific literature that GA_3_ stimulates shoot elongation, the influence of its various concentrations on shoot number and length was investigated in this study. Moreover, the microshoot cultures agitated in liquid media were selected for the experiment from several possible in vitro systems [8,9] due to the possibility of carrying out the elicitation process. Gibberellic acid affected the shoot length but the parameter was not statistically different depending on the GA_3_ concentration used. It was observed that shoot elongation occurred when GA_3_ was included in the culture media [20].

Elicitors are chemicals that can affect the direction of metabolism of plant cells, increasing the accumulation or the de novo production of certain compounds, which are involved in the plant’s defense reaction [21]. The accumulation of secondary metabolites in plant cells is greatly dependent on stressful factors, such as UV-irradiation, light, temperature, drought, heavy metals, and mechanic injury [22]. In addition, biotic stress plays an important role in enhancement of the biosynthetic potential of the plants, e.g., whole cultures of filamentous fungi, supernatants from their culture media and autoclaved biomass or cell wall hydrolysates of fungi, yeast extracts, algae and algae hydrolysates, polysaccharides and oligosaccharides, proteins/enzymes [21]. The use of lichenic compounds to increase the production of bioactive compounds in plant in vitro cultures remains an unexplored area. The present study represents the first examination of the use of (+)-usnic acid as an elicitor of selected phenolic compounds in biomass obtained in vitro from agitated microshoot cultures of *E. alpinum*. Moreover, this is the first report of an elicitation process applied to enhance the secondary metabolites present in this species.

Our previous studies demonstrated that osmotic stress, yeast extract and methyl jasmonate greatly influenced the production of rosmarinic acid (RA), chlorogenic acid and caffeic acid in biomass collected from agitated microshoot cultures of the related species *Eryngium planum* [23]. In the present study, the highest concentration of rosmarinic acid in *E. alpinum* shoots was obtained after 24 h elicitation with 3.125 µM usnic acid (512.69 ± 4.89 mg/100 g DW), which was 2.09-fold greater than in the control sample (Table 3). The yield of rosmarinic acid in *E. planum* shoots positively responded to all studied elicitors, showing a higher concentration of sucrose, yeast extract (YE), and methyl jasmonate (MeJA). However, the highest content of RA was observed after 48 h elicitation with 100 µM of methyl jasmonate (17.73 mg/g DW), which was 5.46-fold higher than in untreated controls [23]. The obtained results indicate that the greatest concentration of chlorogenic acid in *E. alpinum* shoots was obtained after 48 h elicitation with 50 µM usnic acid (43.44 ± 0.33 mg/100 g DW), which was 2.93-fold greater than in the control sample (Table 3). Interestingly, the synthesis of chlorogenic acid (CGA) in *E. planum* shoots was stimulated by almost all studied elicitors, but the highest content of CGA (332 mg/100 g DW) was observed after 20 days treatment with 50 mg/L of sucrose, which was 3.13-fold higher than in untreated controls [23]. The maximum increase of caffeic acid in relation to non-elicited shoots of *E. alpinum* was 1.32-fold (218.38 ± 0.95 mg/100 g DW) after 48 h elicitation with 50 µM usnic acid (Table 3) and 3.75-fold in shoots of *E. planum* after 48 h elicitation with 100 µM of MeJA (30 mg/100 g DW) [23].

Despite some limitations, the enhancement of biosynthesis of phenolic compounds in shoot biomass cultured in vitro by biotic elicitation methods was studied by several authors. Among biotic elicitors in the shoot cultures of many medicinal plants, yeast extract (YE) was the most used. Three days treatment with 200 µM YE resulted in a 1.5-fold increase of RA in microshoot cultures of *Salvia virgata* [24]. Six-week elicitation with 500 mg/L YE enhanced production of RA by 1.62-fold in *Thymus lotocephalus* shoot cultures [25]. The treatment of *Knautia sarajevensis* microshoot cultures with YE (1.0 mg/L or 4.0 mg/L) enhanced the production of rosmarinic acid up to 8.77-fold (153.87 nmol/mL), sinapic acid up to 160.61-fold (28.91 nmol/mL), syringic acid up to 10.65-fold (10.65 nmol/mL), caffeic acid up to 24-fold (47.07 nmol/mL), vanillic acid up to 1.86-fold (56.56 nmol/mL), chlorogenic acid up to 1.79-fold (297.49 nmol/mL), and gallic acid up to 2.78-fold (2.78 nmol/mL) [26]. The very promising results with the use of YE as elicitor were shown for *Schisandra chinensis* agitated microshoot cultures. A large improvement in specific dibenzocyclooctadiene lignan production was reached after the elicitation with 5000 mg/L of YE on the first day of the growth period, and with 1000 and 3000 mg/L on the twentieth day, the lignan production increased to the same degree (about 1.8-fold in comparison to control). Moreover, supplementation with 1000 mg/L YE on the twentieth day of the growth cycle was also applied for the microshoot cultures maintained in Plantform temporary immersion system, where the obtained total content of lignans was equal to 831.6 mg/100 g DW [27].

## 4. Materials and Methods

### 4.1. Plant Material

The voucher specimen has been deposited in the Department of Pharmaceutical Botany and Plant Biotechnology Poznań University of Medical Sciences under the number H-AP-2017-102. Small individuals of *Eryngium alpinum* L. were collected from the Adam Mickiewicz Botanical Garden in Poznań (Poland) in September 2017. The donor plant, which was the source of the explants, grew in the alpinarium (52°25′13.1″N 16°52′44.9″E) under the number A_AG03_001_0030_7591_A0C9. The stem fragments with lateral buds were cut off from first-year young individuals and on the same day used as primary explants to establish and stabilize shoot cultures under controlled conditions [8,9].

### 4.2. Shoot Cultures Initiation

The explants were surface disinfected according to the procedure adopted by Kikowska et al. [8]. Briefly, the young individuals were disinfected with a sodium hypochlorite solution (2.5%) with a few drops of surfactant. The explants were placed in Erlenmeyer flasks with 50 mL solidified MS medium [28] with plant growth regulators (PGRs) (6-benzylaminopurine (BAP; Sigma-Aldrich, Saint Louis, MO, USA), indolile-3-acetic acid (IAA; Sigma-Aldrich, Saint Louis, MO, USA), gibberellic acid (GA_3_; Sigma-Aldrich, Saint Louis, MO, USA) at a concentration of 1.0 mg/L each [8]). The aseptic explants were the source for experiments—in vitro shoot multiplication on agar media. After several series of passages (40 days intervals) and stabilization of the shoot culture on solid media, shoots were collected to establish the shoot culture in a liquid media.

### 4.3. Shoot Cultures Agitated in Liquid Media

Shoots of *E. alpinum* propagated on solid media (see: Section 4.2 Shoot cultures initiation) served as a source of explants to establish shoot cultures in liquid media. The shoots were multiplied via the axillary branching method in liquid MS medium enriched with BAP 1.0 mg/L, IAA 1.0 mg/L and GA_3_ (0.0, 0.5, 1.0, 1.5, 2.0, 2.5, 3.0 mg/L). The 250 mL Erlenmeyer flasks with 20 mL of medium were employed. The cultures were maintained on a rotary shaker (110 rpm). Experiments were repeated two times for 5 explants per treatment.

The cultures were grown under artificial light—55 µmoL/m^2^s (16 h light / 8 h dark photoperiod) and at a temperature of 21 °C ± 2 °C.

### 4.4. Elicitation Process 

In the experiment, the control cultures were non-elicited shoots agitated in liquid medium (0 µM, 0 h). For the elicitation process, the (+)-usnic acid was added to the liquid medium of stabilized shoot culture. Usnic acid was applied in the concentrations: 3.125 µM, 6.25 µM, 12.5 µM, 25 µM, and 50 µM for 12, 24, 48, and 72 h. The (+)-usnic acid was obtained from the compounds’ collection of Pharmacognosy Department, Poznan University of Medical Sciences. The identity, optical rotation and purity were assessed by UV, ^1^H NMR, ^13^C NMR, MS, and polarimetric analysis. After the treatment period, shoots were harvested and content of phenolic compounds (flavonoids and phenolic acids) was estimated by employing HPLC-DAD system. 

### 4.5. HPLC-DAD Analyses 

The *E. alpinum* agitated shoots after the elicitation period (12, 24, 48 and 72 h) were harvested and drained in a sterile lignin. The experimental shoot biomasses were dried at 40 °C for 24 h to a constant weight. The dried and pulverized material (0.5 g dry weight (DW) for each sample) was extracted with methanol (5 mL) for 30 min three times by sonication (Polsonic^®^ 3, Poland). For the analyses of phenolic acids and flavonoids, the validated HPLC-DAD method [29,30] was used. The parameters of analyses were described by us previously [31,32]. The HPLC-DAD system (Merck-Hitachi) and a Purospher RP-18e analytical column (4 × 250 mm, 5 mL; Merck) were employed for the chromatographic quantification. The gradient elution of mobile phase consisting of methanol with 0.5% acetic acid (1:4 *v*/*v*) (A) and methanol (B) was conducted according to the following scheme: 0–20 min, 0% B; 20–35 min, 0–20% B; 35–45 min, 20–30% B; 45–55 min, 30–40% B; 55–60 min, 40–50% B, 60–65 min, 50–75% B; and 65–70 min, 75–100% B, with a hold time of 15 min, at 25 °C. The flow rate was 1 mL/min. The injection volume was 10 µL. The identification of compounds was carried out on the basis of UV-DAD spectra (*λ* = 200–400 nm), Rt (retention time) values and internal reference standard addition method. Quantification was calculated based on comparison peak area measurements to the standard curves. The result was expressed as milligram of studied compound (mg)/100 g dry weight of plant material. The used flavonoid standards accomplished aglycones: isorhamnetin kaempferol, luteolin, quercetin, rhamnetin, and myricetin and glycosides: apigetrin, cynaroside, hyperoside, populnin, quercitrin, rutoside, trifolin, and vitexin (14 compounds). The used phenolic acid standards accomplished: 3,4‒dihydroxyphenylacetic acid, caftaric acid, caffeic acid, chlorogenic acid, *o*‒, *m*‒, *p*‒coumaric acids, cryptochlorogenic acid, ferulic acid, gallic acid, gentisic acid, hydrocaffeic acid, p-hydroxybenzoic acid, isochlorogenic acid, isoferulic acid, protocatechuic acid, rosmarinic acid, salicylic acid, sinapic acid, syringic acid and vanillic acid, and benzoic and cinnamic acids (precursors) (23 compounds). All standards were bought from Sigma-Aldrich, Saint Louis, MO, USA.

### 4.6. Statistical Analysis

The obtained data were analyzed using a one-way analysis of variance (ANOVA) and the statistical significance was determined by Duncan’s POST-HOC test (*p*-value of 0.05). All analyses were conducted using STATISTICA v. 13 (StatSoft, Inc., Kraków, Poland, 2015).

## 5. Conclusions

In vitro shoot cultures of *E. alpinum* (a protected species) may offer a promising source of valuable secondary metabolites as phenolics without harvesting the plant material from the natural environment. Moreover, elicitation with usnic acid may therefore be an effective technique of enhancing the accumulation of phenolic compounds in alpine eryngo agitated shoot cultures. From a general point of view, plant microshoot cultures may serve as a model system for research dealing with accumulation of secondary metabolites.

## Figures and Tables

**Figure 1 molecules-26-05532-f001:**
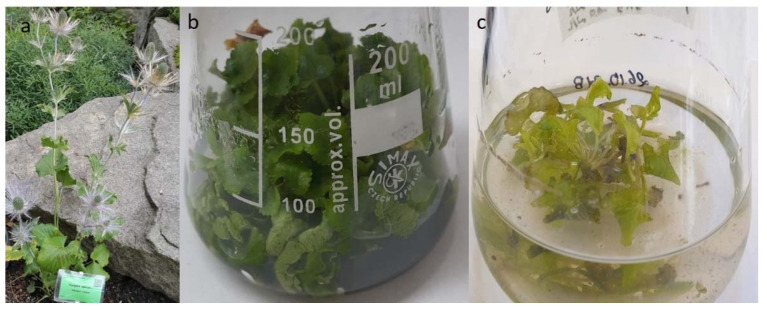
*Eryngium alpinum* L. (**a**) soil-grown plant; (**b**) microshoot culture on solid medium; (**c**) microshoot culture agitated in liquid medium.

**Table 1 molecules-26-05532-t001:** The effect of PGRs on biomass parameters of *Eryngium alpinum* L. microshoot cultures agitated in Murashige and Skoog liquid media.

PGRs (mg/L).			Induction	Shoot No./Explant	Shoot Length
BAP	IAA	GA_3_	[%]	No. ± SD	[cm]
0.0	0.0	0.0	100 ^n,s^	2.67 ± 0.58 ^b^	4.53 ± 0.31 ^b^
1.0	1.0	0.0	100	8.33 ± 1.53 ^a^	5.67 ± 1.04 ^b^
1.0	1.0	1.0	100	9.00 ± 1.00 ^a^	7.60 ± 0.66 ^a^
1.0	1.0	1.5	100	7.66 ± 0.58 ^a^	7.93 ± 0.59 ^a^
1.0	1.0	2.0	100	8.67 ± 0.58 ^a^	8.03 ± 1.17 ^a^
1.0	1.0	2.5	100	8.33 ± 1.53 ^a^	7.77 ± 0.42 ^a^
1.0	1.0	3.0	100	9.33 ± 1.15 ^a^	7.67 ± 0.29 ^a^

Mean values within a column with the same letter are not significantly different at *p* < 0.05 (Duncan’s Multiple Range Test).

**Table 2 molecules-26-05532-t002:** The effect of (+)-usnic acid on selected flavonoid accumulation in *Eryngium alpinum* L. in vitro microshoot biomass.

Treatment		Flavonoids (mg/100 g DW ±SD)			
**Conc.**	Time	Isoquercetin	Robinin	Quercitrin	Total
0	0	82.10 ± 0.62 ^c,d^	15.39 ± 0.23 ^c,d,e^	35.11 ± 0.19 ^d,e,f^	132.60
3.125 µM	12 h	70.97 ± 0.17 ^ef^	18.13 ± 0.44 ^b,c,d^	29.56 ± 0.13 ^I,j^	118.66
3.125 µM	24 h	93.87 ± 1.49 ^b^	37.32 ± 2.59 ^a^	45.88 ± 0.98 ^b^	177.07
3.125 µM	48 h	24.12 ± 5.09 ^j^	9.66 ± 3.84 ^f,g^	42.94 ± 0.34 ^b^	76.72
3.125 µM	72 h	107.17 ± 4.67 ^a^	17.88 ± 0.07 ^b,c,d^	51.08 ± 0.17 ^a^	176.13
6.25 µM	12 h	67.43 ± 4.19 ^f,g^	16.38 ± 2.09 ^b,c,d,e^	28.44 ± 0.46 ^j^	112.25
6.25 µM	24 h	62.38 ± 0.48 ^g,h^	11.97 ± 0.25 ^e,f^	29.10 ± 0.36 ^i,j^	103.45
6.25 µM	48 h	76.04 ± 0.63 ^d,e^	17.36 ± 0.79 ^b,c,d^	33.72 ± 0.75 ^f,g,h^	127.12
6.25 µM	72 h	58.83 ± 0.46 ^h^	6.02 ± 1.11 ^g,h^	32.02 ± 0.25 ^f,g,h,i^	96.87
12.5 µM	12 h	127.54 ± 11.34 ^a^	15.32 ± 6.62 ^c,d,e^	35.13 ± 3.01 ^def^	177.99
12.5 µM	24 h	93.93 ± 0.31 ^b^	19.01 ± 0.51 ^b,c^	32.05 ± 0.03 ^f,g,h,i^	144.99
12.5 µM	48 h	73.81 ± 0.63 ^e,f^	13.93 ± 0.02 ^d,e,f^	34.81 ± 0.05 ^d,e,f,g^	122.55
12.5 µM	72 h	89.55 ± 5.68 ^b,c^	6.12 ± 0.45 ^g,h^	35.19 ± 0.69 ^d,e,f^	130.86
25 µM	12 h	108.37 ± 12.1 ^a^	20.67 ± 6.40 ^b^	38.78 ± 7.21 ^c^	167.82
25 µM	24 h	89.23 ± 0.61 ^b,c^	15.78 ± 0.22 ^b,c,d,e^	31.32 ± 0.06 ^h,I,j^	136.33
25 µM	48 h	85.77 ± 3.63 ^c^	12.34 ± 0.66 ^e,f^	34.08 ± 0.67 ^e,f,g,h^	132.19
25 µM	72 h	62.41 ± 0.33 ^g,h^	5.00 ± 0.12 ^h^	37.28 ± 0.55 ^c,d,e^	104.69
50 µM	12 h	71.93 ± 1.16 ^e,f^	10.35 ± 0.71 ^f,g^	31.59 ± 0.52 ^g,h,i,j^	113.87
50 µM	24 h	31.22 ± 1.53 ^j^	3.77 ± 0.31 ^i^	28.89 ± 0.13 ^i,j^	63.88
50 µM	48 h	106.74 ± 3.35 ^a^	17.63 ± 4.27 ^b,c,d^	38.09 ± 0.84 ^c,d^	162.46
50 µM	72 h	41.36 ± 0.14 ^i^	4.04 ± 0.14 ^i^	34.88 ± 0.04 ^d,e,f,g^	80.28

Mean values within a column with the same letter are not significantly different at *p* < 0.05. (Duncan’s Multiple Range Test).

**Table 3 molecules-26-05532-t003:** The effect of (+)-usnic acid on selected phenolic acid accumulation in *Eryngium alpinum* L. in vitro microshoot biomass.

Treatment		Phenolic Acids (mg/100 g DW ± SD)							
Conc.	Time	Caftaric Acid	3,4-Dihydroxy-phenylacetic Acid	Chlorogenic Acid	Cryptochlorogenic Acid	Isochlorogenic Acid	Caffeic Acid	Rosmarinic Acid	Total
0	0	26.82 ± 0.14 ^g,h^	116.40 ± 2.51 ^h,i^	14.81 ± 0.79 ^e^	55.18 ± 0.47 ^a^	34.52 ± 1.79 ^b^	165.60 ± 0.77 ^d^	245.27 ± 0.89 ^e^	658.60
3.125 µM	12 h	26.34 ± 0.65 ^h^	191.14 ± 10.95 ^b^	6.75 ± 1.03 ^j^	31.89 ± 5.39 ^f,g^	3.81 ± 0.23 ^g^	51.09 ± 6.46 ^j^	204.30 ± 3.38 ^f^	515.32
3.125 µM	24 h	43.32 ± 1.11 ^a^	182.17 ± 4.48 ^b,c^	12.11 ± 1.35 ^f,g^	50.73 ± 6.11 ^a,b^	41.04 ± 0.51 ^a^	200.83 ± 11.93 ^b^	512.69 ± 4.89 ^a^	1042.89
3.125 µM	48 h	7.17 ± 0.20 ^m^	26.95 ± 0.69 ^l^	4.28 ± 0.22 ^k^	14.73 ± 0.04 ^i^	13.45 ± 4.09 ^d,e^	20.98 ± 2.09 ^k^	79.21 ± 4.55 ^i^	166.77
3.125 µM	72 h	32.48 ± 0.88 ^d^	157.38 ± 5.46 ^e,f,g^	14.59 ± 1.71 ^e^	46.80 ± 0.86 ^b,c^	21.13 ± 0.83 ^c^	178.93 ± 7.41 ^c^	361.70 ± 21.23 ^c^	813.01
6.25 µM	12 h	37.31 ± 1.58 ^b^	174.54 ± 9.51 ^c,d^	8.96 ± 0.36 ^h,I,j^	39.53 ± 8.05 ^d,e^	5.93 ± 1.82 ^f,g^	68.47 ± 4.04 ^i^	216.75 ± 12.79 ^f^	551.49
6.25 µM	24 h	27.61 ± 0.37 ^f,g,h^	151.59 ± 1.83 ^e,f,g^	9.98 ± 0.66 ^g,h,i^	41.91 ± 2.00 ^c,d^	6.14 ± 0.79 ^f,g^	100.11 ± 2.31 ^g^	182.47 ± 1.69 ^f^	519.81
6.25 µM	48 h	19.98 ± 0.35 ^j^	64.71 ± 2.03 ^k^	22.13 ± 0.36 ^d^	35.62 ± 0.84 ^e,f^	15.45 ± 0.45 ^d^	79.39 ± 1.79 ^h^	130.49 ± 1.60 ^h^	367.77
6.25 µM	72 h	22.21 ± 1.47 ^i^	125.19 ± 10.97 ^h^	9.91 ± 0.56 ^g,h,i^	27.84 ± 5.32 ^g,h^	10.43 ± 3.12 ^d,e,f^	118.59 ± 2.95 ^f^	263.02 ± 0.76 ^e^	577.19
12.5 µM	12 h	32.84 ± 0.33 ^d^	162.91 ± 2.49 ^d,e,f^	10.39 ± 0.00 ^g,h^	24.21 ± 0.87 ^h^	7.96 ± 3.06 ^e,f,g^	69.30 ± 1.88 ^i^	237.09 ± 15.24 ^e^	544.70
12.5 µM	24 h	35.59 ± 0.80 ^c^	166.64 ± 1.36 ^d,e^	9.64 ± 0.61 ^g,h,i^	47.47 ± 3.23 ^b,c^	13.05 ± 0.36 ^d,e^	126.26 ± 4.74 ^f^	356.09 ± 2.07 ^c^	754.74
12.5 µM	48 h	29.39 ± 0.67 ^e^	171.58 ± 0.78 ^c,d,e^	31.29 ± 1.09 ^b^	0.00 ± 0.00 ^j^	22.22 ± 0.27 ^c^	158.20 ± 1.31 ^d,e^	269.52 ± 1.92 ^e^	682.20
12.5 µM	72 h	28.39 ± 0.35 ^e,f^	161.64 ± 5.23 ^d,e,f,g^	27.71 ± 1.78 ^c^	39.96 ± 5.00 ^d,e^	2.44 ± 0.73 ^g^	175.51 ± 9.26 ^c^	393.93 ± 23.66 ^b,c^	829.58
25 µM	12 h	31.58 ± 1.09 ^d^	205.47 ± 17.02 ^a^	7.73 ± 0.75 ^i,j^	26.19 ± 0.72 ^h^	23.46 ± 10.50 ^c^	94.17 ± 7.44 ^g^	275.12 ± 26.71 ^e^	663.72
25 µM	24 h	34.63 ± 0.43 ^c^	139.77 ± 22.30 ^g^	11.85 ± 3.52 ^g^	49.49 ± 1.18 ^b^	8.44 ± 0.72 ^e,f,g^	119.73 ± 4.95 ^f^	282.17 ± 2.05 ^e^	646.08
25 µM	48 h	28.03 ± 1.13 ^e,f,g^	98.89 ± 3.76 ^j^	32.47 ± 1.48 ^b^	0.00 ± 0.00 ^j^	23.21 ± 1.43 ^c^	164.73 ± 7.20 ^d^	245.76 ± 10.57 ^e^	593.09
25 µM	72 h	22.27 ± 0.17 ^i^	148.21 ± 0.33 ^f,g^	10.32 ± 0.54 ^gh^	34.65 ± 0.83 ^e,f^	15.06 ± 0.45 ^d^	217.78 ± 1.84 ^a^	412.74 ± 1.58 ^b^	861.03
50 µM	12 h	27.34 ± 0.55 ^f,g,h^	160.15 ± 3.41 ^d,e,f,g^	23.89 ± 0.49 ^d^	22.14 ± 0.57 ^h^	5.53 ± 1.38 ^f,g^	118.57 ± 1.69 ^f^	326.16 ± 2.91 ^d^	683.78
50 µM	24 h	18.52 ± 0.78 ^k^	107.26 ± 3.78 ^i,j^	16.59 ± 1.42 ^e^	0.00 ± 0.00 ^j^	10.92 ± 3.59 ^d,e,f^	62.97 ± 2.58 ^i^	185.15 ± 4.29 ^g^	401.41
50 µM	48 h	29.10 ± 0.53 ^e^	125.27 ± 0.95 ^h^	43.44 ± 0.33 ^a^	0.00 ± 0.00 ^j^	33.25 ± 7.86 ^b^	218.38 ± 0.95 ^a^	401.66 ± 22.18 ^b^	851.10
50 µM	72 h	16.29 ± 0.46 ^l^	118.31 ± 2.73 ^h,i^	14.32 ± 3.22 ^e,f^	25.57 ± 4.23 ^h^	11.87 ± 0.33 ^d,e,f^	155.46 ± 2.91 ^e^	263.08 ± 0.76 ^e^	604.90

Mean values within a column with the same letter are not significantly different at *p* < 0.05 (Duncan’s Multiple Range Test).

## Data Availability

The data presented in this study are available on request from the corresponding author.

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
