# Peer review of "Effect of Elicitation with (+)-Usnic Acid on Accumulation of Phenolic Acids and Flavonoids in Agitated Microshoots of Eryngium alpinum L."

_molecules, 2021, doi:10.3390/molecules26185532_

Round 1

Reviewer 1 Report

It appears to me that this article should be in a revised form, and I would like to begin by pointing out that I was not a reviewer of the initial version.

However, this work is of very high quality and relates to a compound of great interest that is too rarely studied (i.e., (+) - usnic acid) which can have many applications if efficient means of production are obtained. Biotechnologies, (the use of elicited micro-shoot) can be an original and efficient approach for this bioproduction (already used industrially), and I therefore strongly support the approaches developed by the authors.

The experiments are perfectly carried out and their descriptions in the materials and methods section adequately described.

The paper is well written. I agree with the conclusions drawn by the authors of these results as well as with the discussion of the latter.

By reading this paper, it is clear to me that the authors (directly or after review) have provided a high-quality work which deserves its publication in Molecules.

Author Response

Dear Reviewer,

on behalf of all the authors who have put an effort to plan and conduct the experiments,  collect the results and prepare this manuscript, I would like to thank you very much for your words of appreciation and opinions in this review. It is also an additional surge of energy for us to plan in-depth research in the field of biotechnology of medicinal plants.

Best regards

Małgorzata Kikowska

Reviewer 2 Report

The present article presents novelty in the focus of the investigation and is characterised by rigorous measurements. Some typos can be found here and there, but they do not disrupt the fluency of the work. Moreover, in the discussion, reporting the comparison with different species AND elicitors may not be extremely relevant due to the presence of two variables; hence the reason of the difference observed may lay in either or both variables. Overall, it is a nice paper with interesting results and possible applications.
Some notes and comments I wanted to share are reported as follows:

-Significant figures should be double checked (e.g. 3.125 uM should be 3.12 uM)
-I would also mention in the main text that normalization by DW has been performed for polyphenols quantitation.
-Is there a possibility that polyphenols may migrate from the biomass to the liquid medium during elicitation? In that case, changing the concentration of usnic acid may change the solubility of the former compounds, affecting the quantitation on methanol extracts. If there are no previous data against this phenomenon, one could try to measure the concentration of polyphenols in liquid media before and after elicitation experiment.
-Double check unit of measure (e.g. L instead of l and mL instead of ml)
-O-, m-, and p- position need to be written in Italic
-Are there any hypothesis on why elicitation with usnic acid affects those polyphenols in particular (e.g. the action on particular biosynthetic pathways or the effects that such compounds have on the "overall well-being" of the shoots)? Including these elements, although difficult, may be interesting for future elucidations.

Author Response

Dear Reviewer, 

we are grateful for your time and constructive comments on our manuscript.

We responded to the suggestions and improved our manuscript.

The minor typos have been corrected.

The additional information (normalization by DW has been performed for polyphenols quantitation) has been added to the experimental part. 

The measurement units (L instead of l and mL instead of ml) have been double checked and replaced correctly. 

O-m-, and p- position has been written in Italic.

Analysis of the content of the tested secondary metabolites in the culture media were performed and did not show the presence of polyphenolic compounds in the liquid media both before and after the elicitation process.

The use of lichenic compound to increase the production of bioactive compounds in plant raw material / in vitro cultures remains an unexplored area. The present study represents the first examination of the employment of (+)-usnic acid as an elicitor of selected phenolic compounds in plant biomass. As we noted in the introduction part "Studies supporting the allelopathic activity of lichen substances on vascular plants have been carried out by several authors, but the adaptive value of this inhibiting effect is still difficult to assess. Due to the allelopathic nature of usnic acid in relation to fungi or plants, an attempt was made to investigate the possibility of playing the role of this lichen acid as an elicitor in relation to plant cells". 

The authors plan to extend this aspect of the study to analyze the biosynthesis pathway of these compounds under the treatment of this particular elicitor.

Best regards 

Małgorzata Kikowska